# Enhancing the Curie Temperature in Cr_2_Ge_2_Te_6_ via Charge Doping: A First-Principles Study

**DOI:** 10.3390/molecules28093893

**Published:** 2023-05-05

**Authors:** Yinlong Hou, Yu Wei, Dan Yang, Ke Wang, Kai Ren, Gang Zhang

**Affiliations:** 1School of Automation, Xi’an University of Posts & Telecommunications, Xi’an 710121, China; 2School of Mechanical and Electronic Engineering, Nanjing Forestry University, Nanjing 210042, China; 3Institute of High Performance Computing, A*STAR, Singapore 138632, Singapore

**Keywords:** magnetic stability, two-dimensional magnets, charge doping, first-principles calculations

## Abstract

In this work, we explore the impacts of charge doping on the magnetism of a Cr_2_Ge_2_Te_6_ monolayer using first-principles calculations. Our results reveal that doping with 0.3 electrons per unit cell can enhance the ferromagnetic exchange constant in a Cr_2_Ge_2_Te_6_ monolayer from 6.874 meV to 10.202 meV, which is accompanied by an increase in the Curie temperature from ~85 K to ~123 K. The enhanced ratio of the Curie temperature is up to 44.96%, even higher than that caused by surface functionalization on monolayer Cr_2_Ge_2_Te_6_, manifesting the effectiveness of charge doping by improving the magnetic stability of 2D magnets. This remarkable enhancement in the ferromagnetic exchange constant and Curie temperature can be attributed to the increase in the magnetic moment on the Te atom, enlarged Cr-Te-Cr bond angle, reduced Cr-Te distance, and the significant increase in super-exchange coupling between Cr and Te atoms. These results demonstrate that charge doping is a promising route to improve the magnetic stability of 2D magnets, which is beneficial to overcome the obstacles in the application of 2D magnets in spintronics.

## 1. Introduction

Nowadays, two-dimensional (2D) magnets, including CrBr_3_ [1], VI_3_ [2,3], Fe_3_GeTe_2_ [4], CrTe_2_ [5,6], MnPS_3_ [7,8], FePS_3_ [9,10], and MnBi_2_Te_4_ [11,12], have drawn substantial attention due to their revolutionary prospects in spintronics [13,14,15,16], since few-layer CrI_3_ and Cr_2_Ge_2_Te_6_ were synthesized via mechanical exfoliation from the bulk [17,18]. Researchers have tried to use 2D magnets in bio-sensing [17], spin filters [18], signal transfer [19], and data storage [20,21]. For instance, Li et al. [17] produced an MXenes-Ti_3_C_2_-CuS nanocomposite exhibiting peroxidase-like activity to sense cholesterol, while Yang et al. [18] designed excellent ultrathin spin filters using the half-metal 2D Cr_2_NO_2_. However, the long-range magnetic order is scarce in 2D magnets at finite temperature due to thermal fluctuations, as suggested by the Mermin–Wagner theorem [22]. At finite temperature, the ordered arrangement of spins would be destroyed by thermal fluctuations, leading to a paramagnetic state in 2D magnets. The critical temperature, where ferromagnetic or antiferromagnetic order transforms into paramagnetic order, is called the Curie or Néel temperature. For 2D magnets, the Curie or Néel temperature is always much lower than the room temperature, such as ~45 K for CrI_3_ [23] and ~89 K for FePS_3_ [10].

To enhance the Curie/Néel temperature of 2D magnets, numerous strategies have been used, such as strain [24,25,26], intercalation [27,28], atom doping [29,30], and surface functionalization [31,32]. For instance, Liu et al. [33] manipulated the ferromagnetism and electronic properties in 2D Cr_2_Ge_2_Te_6_ by strain and found an enhancement in Curie temperature from 59 K to 169 K at 10% strain. Zheng et al. [34] improved the Curie temperature of CrI_3_ to ~211 K through the adsorption of N_2_. Wang et al. [35] found that the adsorption of NO not only increases the exchange coupling between Mn ions in an MnPS_3_ monolayer but also enhances the out-of-plane magnetic anisotropy and induces strong intralayer Dzyaloshinskii–Moriya interaction, contributing to an enhanced Néel temperature. However, the effect of charge doping on the Curie/Néel temperature of a 2D magnet still needs further exploration, although the charge doping is an effective route to switch the interlayer magnetic phase [36,37] and half-metallicity [38] of 2D van der Waals magnets. 

In this paper, we investigate the impacts of charge doping on the magnetic exchange coupling and Curie temperature of a Cr_2_Ge_2_Te_6_ monolayer through first-principles calculations. According to calculations, doping with 0.3 electrons per unit cell can increase the ferromagnetic exchange constant in the Cr_2_Ge_2_Te_6_ monolayer from 6.874 meV to 10.202 meV, resulting in a 44.96% enhancement in Curie temperature. This remarkable increase in Curie temperature reveals that charge doping is a promising route to improve the Curie/Néel temperature and magnetic stability of 2D magnets.

## 2. Results and Discussion

The geometrical structure of the Cr_2_Ge_2_Te_6_ monolayer is shown in Figure 1a, where the red line highlights the optimized unit cell of the Cr_2_Ge_2_Te_6_ monolayer. The lattice constant of an optimized unit cell is *a* = *b* = 6.91 Å, which is slightly larger than the experimental results (6.83 Å) but agreed well with other computational results [39,40]. To identify the dynamical stability of the optimized Cr_2_Ge_2_Te_6_ monolayer, the phonon dispersion is presented in Figure 1b. It is clear that there is no obvious imaginary frequency in the phonon spectrum, suggesting dynamic stability in the optimized Cr_2_Ge_2_Te_6_ monolayer. In addition, we draw the electronic band structure and partial DOS for the Cr_2_Ge_2_Te_6_ monolayer in Figure 1c. In the band structure, a significant spin splitting between the spin-up (red) and spin-down (blue) channels can be observed, indicating a strong magnetism in the Cr_2_Ge_2_Te_6_ monolayer. In the partial DOS, the contributions of Cr, Ge, and Te atoms are represented by the blue, red, and green solid lines, respectively. Obviously, the spin splitting of Cr atoms is much more notable than other atoms, suggesting a dominated contribution from Cr atoms to the net magnetic moments in the Cr_2_Ge_2_Te_6_ monolayer. 

For 2D magnets, the Hamiltonian can be written as:(1)$$H=-\frac{1}{2}\underset{i\neq j}{\sum }J_{ij}S_{i}\cdot S_{j}-\sum_{i,j}\;A_{ij}^{zz}S_{i}^{zz}S_{j}^{zz}$$
where *J_ij_* and $A_{\mathrm{ij}}^{\mathrm{zz}}$ are the isotropic exchange constant and the out-of-plane magnetic anisotropic parameter, respectively. ***S****_i_* = ($S_{i}^{\;x}$, $S_{i}^{\;y}$, $S_{i}^{\;z}$) is the spin vector with an amplitude of ***S***_0_ = 3/2 on the *i*-th magnetic Cr atom. As is known, the out-of-plane magnetic anisotropic energy of the Cr_2_Ge_2_Te_6_ monolayer is only about 3~5% of the nearest-neighbor exchange constant [40,41] and, thus, can be neglected. Hence, the Heisenberg Hamiltonian considering only magnetic exchange coupling can be used to describe the magnetism in the Cr_2_Ge_2_Te_6_ monolayer. The four possible magnetic configurations for the Cr_2_Ge_2_Te_6_ monolayer are presented in Figure 2a–d, including ferromagnetic (FM), antiferromagnetic–Néel (AFM-N), antiferromagnetic–Stripy (AFM-S) and antiferromagnetic–Zigzag (AFM-Z). In Figure 2a–d, the red and gold arrows represent the spin-up and spin-down magnetic moments, respectively. To determine the magnetic ground state, the total energies of the Cr_2_Ge_2_Te_6_ monolayer at four possible magnetic configurations were calculated, and it was found that the energy of the 2 × 2 × 1 Cr_2_Ge_2_Te_6_ supercell in the ferromagnetic state is 372.88 meV, 218.16 meV, and 124.96 meV lower than the AFM-N, AFM-S, and AFM-Z states. Based on the Heisenberg Hamiltonian, the total energies at four possible magnetic phases can be represented as:(2)$$E_{\mathrm{FM}}=E_{0}-12J_{1}S_{0}^{2}-24J_{2}S_{0}^{2}-12J_{3}S_{0}^{2},$$
(3)$$E_{\mathrm{AFM}-N}=E_{0}+12J_{1}S_{0}^{2}-24J_{2}S_{0}^{2}+12J_{3}S_{0}^{2},$$
(4)$$E_{\mathrm{AFM}-S}=E_{0}+4J_{1}S_{0}^{2}+8J_{2}S_{0}^{2}-12J_{3}S_{0}^{2},$$
(5)$$E_{\mathrm{AFM}-Z}=E_{0}-4J_{1}S_{0}^{2}+8J_{2}S_{0}^{2}+12J_{3}S_{0}^{2},$$
where *J*_1_, *J*_2_, and *J*_3_ are the exchange constant for the nearest-neighboring, second-nearest-neighboring, and third-nearest-neighboring coupling. The calculated *J*_1_, *J*_2_, and *J*_3_ are 6.473 meV, −0.207 meV, and 0.432 meV, respectively, which is consistent with the results in Zhang’s paper [39]. It is apparent that the second- (*J*_2_) and third- (*J*_3_) nearest-neighboring exchange constants are much smaller than the nearest-neighboring exchange constant (*J*_1_), at only ~3% and ~6.7% of *J*_1_, respectively. Hence, only the nearest-neighbor interaction was considered in this work, which was also adopted in previous studies [33,39,41,42]. To calculate the isotropic exchange constant (*J*_1_) of the nearest-neighbor magnetic coupling, two magnetic configurations, including FM and AFM-N of the Cr_2_Ge_2_Te_6_ unit cell, are sufficient. Therefore, *J*_1_ can be calculated by:(6)$$J_{1}=-\left(E_{\mathrm{FM}}-E_{\mathrm{AFM}}\right)/\left(6S_{0}^{2}\right)$$
where *E*_FM_ and *E*_AFM_ are the total energies for the unit cell of the Cr_2_Ge_2_Te_6_ monolayer in FM and AFM-N states, respectively. 

In the unit cell of the pristine Cr_2_Ge_2_Te_6_ monolayer, *J*_1_ was calculated as 6.874 meV, in close agreement with Wu’s and Wang’s results [39,43]. It is clear that the magnetic exchange constant is positive, revealing a ferromagnetic ground state. This ferromagnetic ground state can also be verified by the spin charge density in Figure 2e. In Figure 2e, the red (spin up) and gold (spin down) isosurfaces of spin charge density are separated by ±0.003 e/Å^3^. Meanwhile, we find that the majority spin occurs on Cr atoms while that on Te atoms is slight, consistent with the partial DOS in Figure 1c. However, why do Te atoms, presumably a doubly negative anion, bear a slight magnetic moment? To answer this question, we calculated the electron density distribution (EDD) and the electron localization function (ELF) of the pristine Cr_2_Ge_2_Te_6_ monolayer, and the results are presented in Figure 3a,b. In Figure 3a, the isosurfaces of EDD are 0.055 e/Å^3^, while the value of ELF within a range of 0~1 is represented by the color from blue to red (B-G-R). In the unit cell of the Cr_2_Ge_2_Te_6_ monolayer, each Te atom is surrounded by two Cr atom and one Ge atom. Interestingly, obvious electron transfer between Ge and Te atoms can be observed in EDD, while the electrons around Cr atoms show a remarkable localization. Electrons accumulate around Te atoms and dissipate around Ge atoms, resulting in small magnetic moments on Te and Ge atoms. The magnetic moment is about −0.1 $\mu_{B}$ per Te atom, while that is 0.047 $\mu_{B}$ on every Ge atom. In Figure 2e, the isosurfaces of spin charge density are separated by ±0.003 e/Å^3^, leading to the omission of weak magnetism on Ge atoms. If the difference between isosurfaces in spin charge density is set as ±0.002 e/Å^3^, the spin charge density of the pristine Cr_2_Ge_2_Te_6_ monolayer is shown in Figure 3c, where tiny positive magnetic moment on Ge atoms can also be seen. The underlying characteristics of bonding properties around Te atoms in the Cr_2_Ge_2_Te_6_ monolayer is revealed by the value of ELF in Figure 3b. If the value of ELF is below 0.5, the bonding would be regarded as ionic. If this value is in a range of 0.5~1, covalent bonding is suggested. In Figure 3b, the value of ELF between Te and Cr atoms is close to zero, indicating strong ionicity, while that between Te and Ge atoms is slightly smaller than 0.5, also revealing ionic bonding and confirming the electron transfer between Ge and Te atoms. 

Subsequently, *J*_1_ in charge-doped Cr_2_Ge_2_Te_6_ monolayers was also calculated through a similar process, and the results are shown in Figure 4a. As mentioned above, the out-of-plane magnetic anisotropic energy of the Cr_2_Ge_2_Te_6_ monolayer is only about 3~5% of the nearest-neighboring exchange constant [40,41] and, thus, it is always neglected in the estimation of Curie temperature. However, does charge doping increase the magnetic anisotropy in the Cr_2_Ge_2_Te_6_ monolayer or not? For magnetic materials, magnetic anisotropic energy (MAE) is one of the most important properties of magnetic materials [44,45], especially when magnets are applied in nano magnetic memory devices [46]. We calculated the magnetic anisotropic energies in Cr_2_Ge_2_Te_6_ monolayers, with and without charge doping. The out-of-plane MAE is only 0.23 meV in the pristine Cr_2_Ge_2_Te_6_ monolayer, which is ~3.3% of the magnetic exchange constant (6.874 meV) and consistent with the results of Xu et al. [41] and Zhang et al. [39]. After 0.3 electron doping, MAE transfers from the out-of-plane to the in-plane direction and increases to 0.59 meV, keeping only ~5.7% of the magnetic exchange constant (10.202 meV). Meanwhile, 0.3-hole doping results in an out-of-plane MAE of 0.42 meV. It can be found that MAE in the Cr_2_Ge_2_Te_6_ monolayer is always one order of magnitude smaller than the nearest-neighboring magnetic exchange constant, regardless of whether charge doping is applied or not. Therefore, we did not pay too much attention to it in this work. 

To estimate the magnetic stability of the Cr_2_Ge_2_Te_6_ monolayer before and after charge doping, Monte Carlo simulations were employed to obtain the Curie temperature. The temperature-dependent normalized *S*_0_ before and after charge doping is presented in Figure 4b, while the gained Tc of Cr_2_Ge_2_Te_6_ monolayers with and without charge doping is listed in Table 1. The gained Curie temperature of the pristine Cr_2_Ge_2_Te_6_ monolayer is ~85 K via Monte Carlo simulation. It is evident that 0.1 electron (0.1 e) doping can improve Tc to 97.97 K, while 0.1-hole (0.1 h) doping decreases Tc slightly. This trend was also observed in the experiment of Chen et al. [47] To evaluate the improvement in T_c_ quantitatively, the enhanced ratio is defined as:(7)$$\mathrm{Ehanced}\;\mathrm{Ratio}=\frac{T_{c}-T_{c}^{\mathrm{pristine}}}{T_{c}^{\mathrm{pristine}}}\times 100\%$$

Figure 4c gives the enhanced ratios of T_c_. The largest enhanced ratio of 44.96% results by doping with 0.3 electrons (0.3 e system), while the largest reduced ratio is 9.41% and induced by 0.1 h doping. This improvement in T_c_ is even more remarkable than that caused by NO molecular adsorption (38%) [48], manifesting the significant effect of charge doping on magnetism and paving a novel route to reinforce the long-range magnetic order in magnetic materials. Furthermore, although the adsorption of NO could induce a remarkable enhancement (38%) in the Curie temperature of the Cr_2_Ge_2_Te_6_ monolayer, Wang et al. [43] proved that the interlayer vibrational modes induced by NO adsorption shorten the magnon relaxation time by ~12.7%. In addition, atom doping is also an effective method to manipulate the physical properties of 2D materials, such as band structure [49], magnetism [50,51,52], mechanical performance [53], and thermal conductivity [54]. For the Cr_2_Ge_2_Te_6_ monolayer, Zhou et al. [55] found that Cr1.9Mn0.1Ge2Te6 displays a decent figure of merit (0.63) at 833 K, a 2-fold value as compared to that of the undoped sample in the same direction and at the same temperature, because Mn doping is favorable to improve the electrical properties of Cr_2_Ge_2_Te_6_. In 2020, Zhang et al. [56] found the substitutional doping of Te with Se atoms can induce a strong Dzyaloshinskii–Moriya interaction and nanometric skyrmions. However, the nearest-neighboring magnetic exchange constant in the Cr_2_Ge_2_Te_6_ monolayer was decreased from 6.14 to 3.267 meV by the substitutional doping of Te with Se in Zhang’s study. Meanwhile, they found some noticeable imaginary frequencies in the phonon dispersion of the Se-doped Cr_2_Ge_2_Te_6_ monolayer, indicating a weak dynamical instability. 

Li et al. [57] reported that charge doping also significantly affects the phonon dispersion of 2D materials, such as graphene-like borophene. To identify the stability of the charge-doped Cr_2_Ge_2_Te_6_ monolayer, we present the phonon spectra of 0.3 e and 0.3 h systems in Figure 5. Obviously, there is no significant imaginary frequency in the phonon spectra of both 0.3 e and 0.3 h systems, revealing that charge doping does not destroy the dynamical stability of the Cr_2_Ge_2_Te_6_ monolayer. Meanwhile, we also calculated the formation energies (*E_f_*) by $E_{f}=E_{tot}-6\mu_{\mathrm{Te}}-2\mu_{\mathrm{Ge}}-2\mu_{\mathrm{Cr}}$, as shown in Table 1. *E_tot_* is the total energy of the Cr_2_Ge_2_Te_6_ system in the ferromagnetic ground state, while *μ*_Cr_, *μ*_Ge,_ and *μ*_Te_ are the chemical potential for Cr, Ge, and Te atoms, respectively. It can be observed in Table 1 that all formation energies are negative, suggesting that these systems in our manuscript are thermodynamically stable. In Table 1, the 0.3 e system owns the lowest formation energy, so its ferromagnetic order possesses the strongest thermodynamic stability, consistent with the largest Curie temperature in the 0.3 e system. In addition, we calculated the band structures of the doped Cr_2_Ge_2_Te_6_ monolayer with 0.3 electrons, as shown in Figure 6a, where the spin-up and spin-down channels are represented by the red and blue lines, respectively. Obviously, the 0.3 electron-doped Cr_2_Ge_2_Te_6_ monolayer (0.3 e system) shows a half-metallic behavior. To test the robustness of this half-metallic state in the doped system, we also calculated the band structures of the 0.3 e system under 2% biaxial lattice expansion and compression in Figure 6b,c. When biaxial strain is applied, it can be found that the conduction band minimum (CBM) remains almost unchanged, while the valence band maximum (VBM) reduces with expansion and elevates with compression, but the half-metallic state in the doped system is not affected. This phenomenon indicates that the half-metallic state in the doped system is robust. In experiments, charge doping can be realized by gate doping, where only one gate electrode (top or bottom) is required. For Cr_2_Ge_2_Te_6_, Verzhbitskiy et al. [58] observed ferromagnetism in a Cr_2_Ge_2_Te_6_ film at temperatures up to 200 K when the gate voltage was set as 3.9 V. In their study, the injected electron density was estimated to be ~4 × 10^14^ cm^−2^ for V_G_ = 3.9 V, corresponding to 1.9 electron doping per unit cell. Therefore, the required gate voltage of 0.6 V should be applied to achieve a doping level of 0.3 electrons in our manuscript.

In the Cr_2_Ge_2_Te_6_ monolayer, the ferromagnetic ground state is caused by the competition between the antiferromagnetic direct exchange interaction (DEI) and the ferromagnetic super-exchange coupling (SEC), as presented in Figure 7a. In Figure 2e and Figure 3c, it is evident that the spin orientation on the Cr atom is antiparallel to that on Te atoms, disclosing a weak antiferromagnetic interaction between them. Although this antiferromagnetic interaction is very weak, it plays a fundamental role in stabilizing the ferromagnetic order in 2D Cr_2_Ge_2_Te_6_ and resisting thermal fluctuation, as per the Goodenough–Kanamori rule [59,60]. In the Goodenough–Kanamori rule, the magnetic exchange constant is a result of the competition between super-exchange and direct exchange coupling. The sign of the magnetic exchange constant depends heavily on the metal–ligand–metal angle. If the angle is greater than 90°, the magnetic exchange constant is dominated by the super-exchange coupling and its sign is positive. If not, direct exchange coupling dominates the magnetic exchange coupling, leading to a negative magnetic exchange constant. This rule is still applicable in newly discovered two-dimensional magnets, such as Cr_2_Ge_2_Te_6_ [48], CrSTe [61], CrI_3_ [62], and MXenes [63].

In Cr_2_Ge_2_Te_6_, Cr^3+^ has the d^3^ high-spin configuration occupying the t_2g_ orbital, as shown in Figure 7b,c. In DEI, direct t_2g_ orbital overlapping between the intralayer adjacent Cr^3+^ ions accounts for the negative exchange constant, which is dominated primarily by the distance between nearest-neighbor Cr^3+^ ions. The Cr-Te-Cr SEC is caused by the overlap of t_2g_ orbit on Cr^3+^ ion and *p* orbit on Te atoms, depending on the magnetic moment on the Te atom, Cr-Te bond length, and Cr-Te-Cr bond angle. Hence, the magnetic moment on the Te atom, the Cr-Cr distance, the Cr-Te bond length, and the Cr-Te-Cr bond angle in the pristine and doped Cr_2_Ge_2_Te_6_ monolayer are exhibited in Table 1 to explore the origin of the enhancement in T_c_. In both the pristine and charge-doped Cr_2_Ge_2_Te_6_ systems, the Cr-Te-Cr bond angles are greater than 90° (as shown in Table 1), and the sign of all magnetic exchange constants is positive, while the Cr-Cr distance is almost unchanged with doping, indicating the insensitivity of antiferromagnetic DEI to charge doping. Therefore, the magnetic stability of these systems is determined by SEC. For the Cr-Te-Cr SEC, although the antiferromagnetic coupling between Te and Cr atoms is weak due to the tiny negative magnetic moment on Te atoms, this coupling is still essential. Therefore, it can be concluded that the weak antiferromagnetic coupling plays a key role in magnetic stability against thermal fluctuation. 

For evaluating the SEC strength between Te and Cr atoms (C_Cr-Te_) quantitatively, the covalency is defined as: (8)$$C_{\mathrm{Cr}-\mathrm{Te}}=-\left|BC^{\mathrm{Cr}}-BC^{\mathrm{Te}}\right|$$
where *BC* represents band center and can be calculated by
(9)$$BC=\frac{\int_{\varepsilon_{0}}^{\varepsilon_{1}}\varepsilon g(\varepsilon )d\varepsilon }{\int_{\varepsilon_{0}}^{\varepsilon_{1}}g(\varepsilon )d\varepsilon }$$
with the PDOS g(ε) and band energy *ε*. The partial DOSs of Cr and Te atoms in the doped Cr_2_Ge_2_Te_6_ monolayer are presented in Figure 8, which were obtained via Gaussian smearing with a small SIGMA and highly consistent with those in previous studies [33,40,64,65]. Subsequently, the obtained covalency C_Cr-Te_ is given in Table 1. According to Equations (4) and (5), a larger C_Cr-Te_ suggests a larger orbital overlap between two atoms, addressing a stronger coupling. In Table 1, the largest C_Cr-Te_ is −1.034 eV in the 0.3 e system revealing the strongest Cr-Te-Cr SEC as the strength of antiferromagnetic DEI remained unchanged, resulting in the largest enhanced ration on the positive *J*_ij_ and the Curie temperature T_c_. The smallest C_Cr-Te_ of −1.130 eV is in the 0.1 h system due to the longest Cr-Te bond length and the smallest Cr-Te-Cr bond angle, leading to a slight decrease in *J*_ij_ and T_c_. In addition, C_Cr-Te_, in both the 0.2 h and 0.3 h systems, is larger than that in the 0.1 h system but still smaller than that in pristine the Cr_2_Ge_2_Te_6_ monolayer, also accounting for a slight drop in *J*_ij_ and T_c_. 

## 3. Computational Method

All first-principles calculations were implemented by the *Vienna* ab initio *simulation package* (VASP) [66,67]. Early in 2014, Pham et al. [68] demonstrated the significance of the Fock exchange in accurately describing dopant geometry and the super-exchange interaction in doped systems. Generally, the results obtained using the hybrid functionals, including the geometrical and magnetic exchange coupling, are closer to the experimental results [69,70], compared with the Perdew–Burke–Ernzerhof (PBE) method of general gradient approximation, because of the different nature of bonding in the PBE and hybrid functionals. In 2020, Joshi et al. [71] proved that the difference between the magnetic exchange couplings obtained using the PBE and hybrid functionals is only about ~2%. However, the consumption of hybrid functionals in memory and computational time is much larger than the PBE method, especially for magnetic materials. To balance the accuracy and required resources of first-principles calculations, we chose the PBE of general gradient approximation with spin polarization as an exchange-correlation functional [72,73], as adopted in previous studies [74,75,76,77]. Meanwhile, the Hubbard ‘*U*’ should be employed to describe the tight-binding *d*-orbital electrons in Cr atoms [78,79]. Gong et al. [80] and Kang et al. [81] reported that the value of “*U*” for Cr atoms in Cr_2_Ge_2_Te_6_ should be selected within a range of 0.2~1.7 eV due to the sensitivity of the band gap and magnetic phase. In Kang et al.’s study, the Cr_2_Ge_2_Te_6_ monolayer behaved as a semiconductor with finite band gap when *U* = 0, while when *U* = 3 eV, it was half-metallic. Meanwhile, Gong et al. found that the bulk Cr_2_Ge_2_Te_6_ becomes the in-plane anisotropic as *U* < 0.2 eV; when *U* > 1.7 eV, the interlayer exchange coupling in the *R*3 phase becomes antiferromagnetic, which is contrary to the experimental results. In our calculations, the value of “*U*” was set as 1 eV. The convergence limits of energy and the Hellmann–Feynman force were set as 10^−8^ eV and 10^−3^ eV/Å, respectively, when the geometrical structure was optimized using 500 eV cutoff energy and 5 × 5 × 1 Monkhorst–Pack (MP) grid. A denser MP grid of 9 × 9 × 1 was used in self-consistent calculations for better results of density of states (DOSs). To suppress the non-physical interaction between adjacent layers, a 20 Å vacuum space was imposed along the out-of-plane direction. To calculate phonon dispersion, a 21 × 21 × 1 MP grid was used, and then phonon frequencies were obtained via a PHONOPY code based on density function perturbation theory (DFPT) [82]. To obtain the Curie temperature accurately, Monte Carlo simulations were employed and implemented via MCSOLVER code [83,84,85]. In all Monte Carlo simulations, a 32 × 32 × 1 supercell was used, and the temperature increased from 2 to 300 K, with a step of ~5 K. In every simulation, 10^7^ loops were implemented with the Metroplis algorithm to achieve equilibrium at each examined temperature. During all simulations, there was no external magnetic field applied. In addition, when the charge doping was introduced, the homogeneous background charge was assumed to compensate for the charge distribution and keep the overall charge neutrality.

## 4. Conclusions

In summary, we investigated the influence of charge doping on the magnetism of a Cr_2_Ge_2_Te_6_ monolayer through first-principles calculations using the framework of density functional theory. According to calculations, we found that the 0.3 electron doping can elevate the Curie temperature of the Cr_2_Ge_2_Te_6_ monolayer from ~85 K to ~123 K, corresponding to an enhanced ratio of 44.96%. This remarkable enhanced ratio can be interpreted by the increased magnetic moment in Te atoms, reduced Cr-Te bond length, an increase in Cr-Te-Cr bond angle, and the strengthening of Cr-Te-Cr SEC. The strength of Cr-Te-Cr SEC is characterized by the covalency between Cr and Te atoms quantitatively. The covalency between Cr and Te atoms in the 0.3 e system is up to −1.034, much larger than that in the pristine Cr_2_Ge_2_Te_6_ monolayer. These results suggest that charge doping can tailor the magnetic properties of 2D magnets effectively and can be used to improve the magnetic stability of 2D magnets.

## Figures and Tables

**Figure 1 molecules-28-03893-f001:**
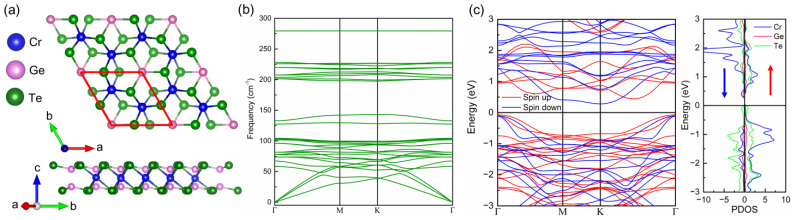
Geometrical structure (**a**), phonon spectrum (**b**), band structure, and partial density of states (DOS) (**c**) of pristine Cr_2_Ge_2_Te_6_ monolayer.

**Figure 2 molecules-28-03893-f002:**
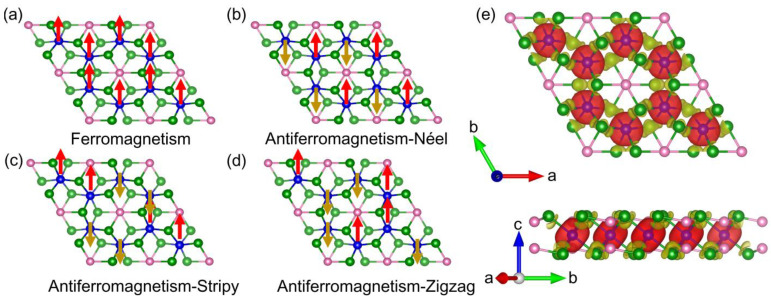
(**a**–**d**) Diagram of four possible magnetic configurations of Cr_2_Ge_2_Te_6_ monolayer, including ferromagnetic (FM), antiferromagnetic–Néel (AFM-N), antiferromagnetic–Stripy (AFM-S), and antiferromagnetic–Zigzag (AFM-Z). (**e**) The spin charge density of pristine Cr_2_Ge_2_Te_6_ monolayer.

**Figure 3 molecules-28-03893-f003:**
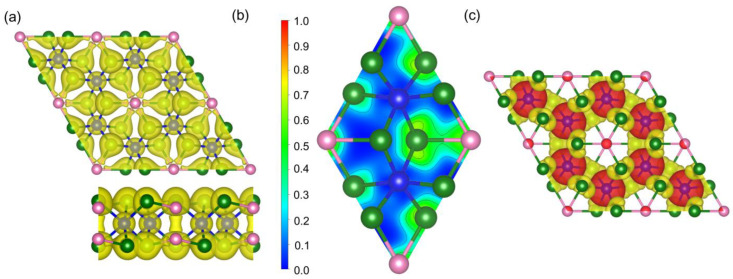
Electron density (**a**) and two-dimensional electron localization function (ELF) (**b**) and spin charge density (**c**) for pristine Cr_2_Ge_2_Te_6_. In (**c**), the isosurfaces of spin charge density are separated by ±0.002 e/Å^3^.

**Figure 4 molecules-28-03893-f004:**
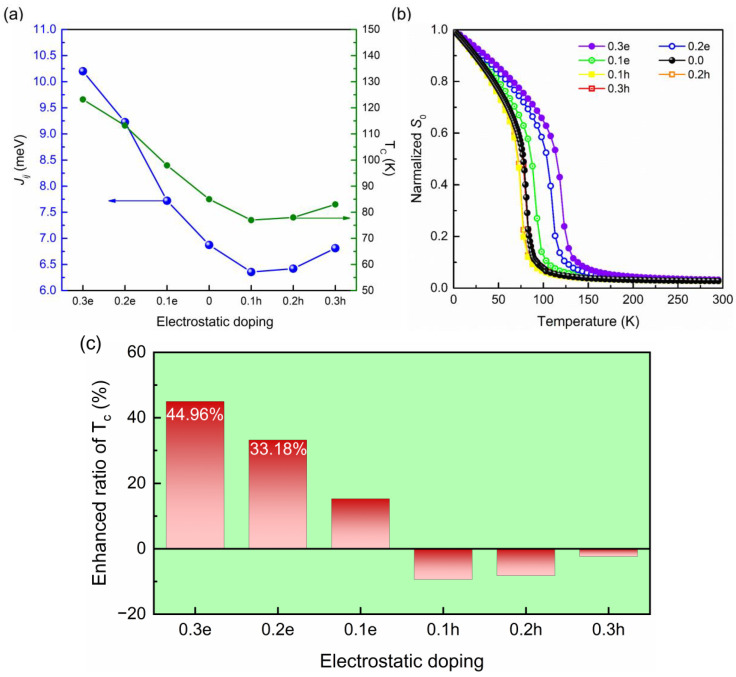
(**a**) Magnetic exchange constant (*J_ij_*) and Curie temperature (T_c_), (**b**) temperature-dependent normalized *S*_0_ before and after charge doping, and (**c**) enhanced ratio of Curie temperature after charge doping.

**Figure 5 molecules-28-03893-f005:**
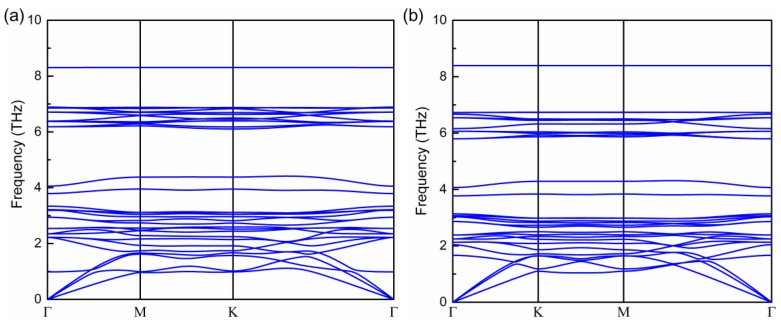
Phonon spectrum for 0.3 electron-doped Cr_2_Ge_2_Te_6_ monolayer (**a**) and 0.3 hole-doped Cr_2_Ge_2_Te_6_monolayer (**b**).

**Figure 6 molecules-28-03893-f006:**
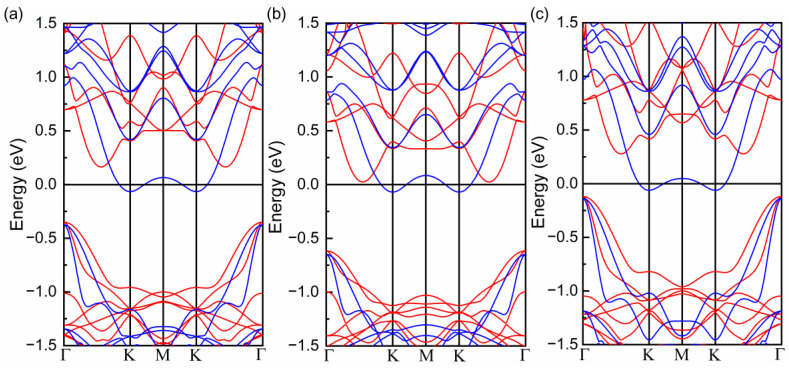
(**a**) Band structure of 0.3 electron-doped Cr_2_Ge_2_Te_6_ monolayer (0.3 e system), and band structures of 0.3 e system under 2% biaxial expansion (**b**) and compression (**c**). The spin-up and spin-down channels are represented by the red and blue lines, respectively.

**Figure 7 molecules-28-03893-f007:**
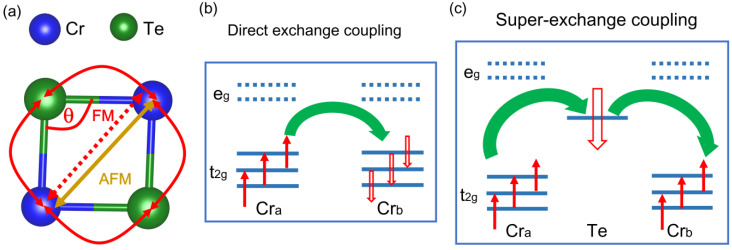
Schematic diagram of the competition between the direct (Cr-Cr) exchange interaction and the super-exchange (Cr-Te-Cr) coupling (**a**), direct exchange mechanism (**b**), and super-exchange mechanism (**c**).

**Figure 8 molecules-28-03893-f008:**
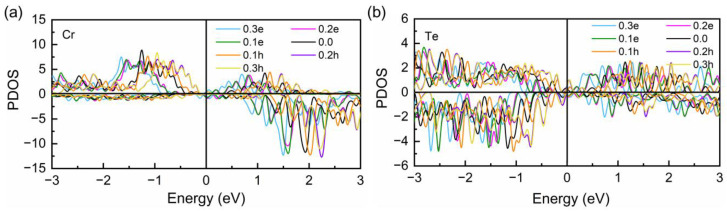
Partial DOS of Cr (**a**) and Te (**b**) atoms in Cr_2_Ge_2_Te_6_ monolayer with and without charge doping.

**Table 1 molecules-28-03893-t001:** The average magnetic moment on Te atoms (M_Te_), the length of Cr-Te bond (L_Cr-Te_), the Te-Cr-Te bond angle (**θ _Cr-Te-Cr_**), the Cr-Cr distance (L_Cr-Cr_), the covalency between Cr and Te atoms (C_Cr-Te_), formation energy (*E_f_*), and the Curie temperature (T_c_) in Cr_2_Ge_2_Te_6_ monolayer with and without charge doping.

System	M_Te_ (μ_B_)	L_Cr-Te_ (Å)	θ_Cr-Te-Cr_ (º)	L_Cr-Cr_ (Å)	C_Cr-Te_ (eV)	*E_f_* (eV)	T_c_ (K)
0.3 e doping	−0.088	2.772	92.053	3.989	−1.034	−14.62	123
0.2 e doping	−0.090	2.772	92.048	3.989	−1.082	−14.55	113
0.1 e doping	−0.086	2.774	91.953	3.989	−1.092	−14.43	98
pristine	−0.083	2.777	91.849	3.989	−1.101	−14.28	85
0.1 h doping	−0.089	2.779	91.744	3.989	−1.130	−14.05	77
0.2 h doping	−0.087	2.781	91.653	3.989	−1.120	−13.78	78
0.3 h doping	−0.089	2.784	91.535	3.989	−1.106	−13.46	83

## Data Availability

The data presented in this study are available upon request from the corresponding author.

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
