# Peer review of "Enhancing the Curie Temperature in Cr2Ge2Te6 via Charge Doping: A First-Principles Study"

_molecules, 2023, doi:10.3390/molecules28093893_

Round 1

Reviewer 1 Report

This submission explores the magnetic behavior of two-dimensional Cr2Ge2Te6 by presenting exciting band structure, phonon dispersion, and temperature dependant Monte Carlo calculations. Given that 2D magnetic materials hold great promise in spintronics and data storage applications, this paper can be published after addressing the following problems:

Can the values of the Curie temperatures be stated in the abstract?

Regarding the discussions in the second paragraph of page 6, can the authors elaborate why Te, presumably a doubly negative anion, should bear any magnetic moment? Can bond ionicity vs. covalency explain this point?

In the same place as above, can the authors explain in detail how the weak antiferromagnetic coupling stabilizes against thermal fluctuations? The cited Ref. 57 is archaic and does not reference these newly discovered materials.

Some details of the Monte Carlo simulation are missing: How large was the simulation box, were periodic boundary conditions assumed, and what was the strength of the applied external magnetic field (if any)? How many cycles were necessary to achieve equilibrium at each examined temperature? How large were temperature intervals?

It would be beneficial for the readers that the authors note that higher-level functionals that account for the electronic correlations are essential not only for obtaining accurate band structures and magnetic moments but also critical for predicting correct geometries, especially in doped systems:

Pham et al. Critical role of Fock exchange in characterizing dopant geometry and magnetic interaction in magnetic semiconductors, Phys. Rev. B 89, 155110: https://doi.org/10.1103/PhysRevB.89.155110

Adding this point will help address a common misconception in the community.

The VASP LDA+U methods, specified by the LDAUTYPE key, are based on one of these two references. None were cited. If the authors used customized code, a note should be included in the paper:

A. I. Liechtenstein, V. I. Anisimov, and J. Zaanen, Phys. Rev. B 52, R5467 (1995).

S. L. Dudarev, G. A. Botton, S. Y. Savrasov, C. J. Humphreys, and A. P. Sutton, Phys. Rev. B 57, 1505 (1998).

Author Response

Response to Referee 1

We thank the referee for her/his insightful suggestions, which have been very helpful for us to enhance the quality of our manuscript. We have revised the manuscript according to the referee’s suggestions. The changes in manuscript have been highlighted in the revised version. Below is our response to the comments.

Comment 1. Can the values of the Curie temperatures be stated in the abstract?

Author Response: We thank the referee for this important comment. In the revised manuscript, we have stated the values of Curie temperatures in abstract.

Abstract. “Our results reveal doping with 0.3 electrons per unit cell can enhance the ferromagnetic exchange constant in Cr2Ge2Te6 monolayer from 6.874 meV to 10.202 meV, which is accompanied by the increase of the Curie temperature from ~85 K to ~123 K. The enhanced ratio of the Curie temperature is up to 44.96% even higher than that caused by surface functionalization on monolayer Cr2Ge2Te6, manifesting the effectiveness of charge doping in improving the magnetic stability of 2D magnets.”

Comment 2. Regarding the discussions in the second paragraph of page 6, can the authors elaborate why Te, presumably a doubly negative anion, should bear any magnetic moment? Can bond ionicity vs. covalency explain this point?

Author Response: We thank the referee for this important comment. You are right that the magnetic moment on Te atoms should be owed to the electron transfer and bond ionicity, which can be identified by the electron density and electron localization function (ELF) in Figure A. In Figure A, the isosurface of electron density distribution is 0.055 e/Å3, while the value of ELF within the range of 0~1 is represented by the colour from blue to red (B-G-R). In Cr2Ge2Te6 monolayer, each Te atoms is surrounded by two Cr and one Ge atoms. Obvious electron transfer between Ge and Te atoms can be observed in electron density distribution, while the electrons around Cr atoms show a remarkable localization. Electrons accumulate around Te atoms and dissipate around Ge atoms, resulting in small magnetic moments on Te and Ge atoms. The magnetic moment is about -0.1  per Te atom, while that is 0.047  on every Ge atom. In Figure 2 in main manuscript, the isosurface of spin charge density is 0.003 e/Å3, leading to the omission of weak magnetism on Ge atom. If the isosurface of spin charge density is set as 0.002 e/Å3, the spin charge density of pristine Cr2Ge2Te6 monolayer is shown in Figure A(c) where tiny positive magnetic moment on Ge atoms can also be seen. The underlying characteristics of bonding properties around Te atoms in Cr2Ge2Te6 monolayer is revealed by the value of ELF in Figure A(b). If the value of ELF is below 0.5, the bonding would be regarded as ionic. If this value is in the range of 0.5~1, covalent bonding is suggested. In Figure A(b), the value of ELF between Te and Cr atoms is closed to zero, indicating strong ionicity, while that between Te and Ge atoms is slightly smaller than 0.5, also revealing ionic bonding and confirming the electron transfer between Ge and Te atoms.

Figure A. Electron density (a) and two-dimensional electron localization function (ELF) (b), and spin charge density (c) for pristine Cr2Ge2Te6.

In the revised manuscript, we have added Figure A as Figure 3 and added following sentences to address this point.

Page 7. “However, why Te atoms, presumably a doubly negative anion, bear slight magnetic moment? To answer this question, we calculated the electron density distribution (EDD) and the electron localization function (ELF) of pristine Cr2Ge2Te6 monolayer, and presented the results in Figure 3a-3b. In Figure 3a, the isosurfaces of EDD is 0.055 e/Å3, while the value of ELF within the range of 0~1 is represented by the color from blue to red (B-G-R). In the unit cell of Cr2Ge2Te6 monolayer, each Te atoms is surrounded by two Cr and one Ge atoms. Interestingly, obvious electron transfer between Ge and Te atoms can be observed in EDD, while the electrons around Cr atoms show a remarkable localization. Electrons accumulate around Te atoms and dissipate around Ge atoms, resulting in small magnetic moments on Te and Ge atoms. The magnetic moment is about -0.1  per Te atom, while that is 0.047  on every Ge atom. In Figure 2e, the isosurfaces of spin charge density are separated by 0.003 e/Å3, leading to the omission of weak magnetism on Ge atom. If the difference between isosurfaces in spin charge density is set as 0.002 e/Å3, the spin charge density of pristine Cr2Ge2Te6 monolayer is shown in Figure 3c where tiny positive magnetic moment on Ge atoms can also be seen. The underlying characteristics of bonding properties around Te atoms in Cr2Ge2Te6 monolayer is revealed by the value of ELF in Figure 3b. If the value of ELF is below 0.5, the bonding would be regarded as ionic. If this value is in the range of 0.5~1, covalent bonding is suggested. In Figure 3b, the value of ELF between Te and Cr atoms is closed to zero, indicating strong ionicity, while that between Te and Ge atoms is slightly smaller than 0.5, also revealing ionic bonding and confirming the electron transfer between Ge and Te atoms.”

Comment 3. In the same place as above, can the authors explain in detail how the weak antiferromagnetic coupling stabilizes against thermal fluctuations? The cited Ref. 57 is archaic and does not reference these newly discovered materials.

Author Response: We thank the referee for this important comment. Although Ref.[57] is archaic, it is a very important work in explaining the theory of magnetic exchange coupling in magnetic materials and a critical component of the famous Goodenough-Kanamori rule. In the Goodenough-Kanamori rule, the magnetic exchange constant is result of the competition between super-exchange and direct exchange coupling. The sign of the magnetic exchange constant depends heavily on the metal-ligand-metal angle. If the angle is greater than 90 degrees, the magnetic exchange constant is dominated by the super-exchange coupling and its sigh is positive. If not, direct exchange coupling dominates the magnetic exchange coupling leading to a negative magnetic exchange constant. This rule is still applicable in newly discovered two-dimensional magnets, such as Cr2Ge2Te6 (J. Mater. Chem. C, 2019, 7, 5084), CrSTe (PHYSICAL REVIEW B 102, 094425 (2020)), CrI3 (J. Phys. Chem. C 2020, 124, 7585-7590), and MXenes (ACS Nano 2019, 13, 2831-2839).

In Cr2Ge2Te6, the short range Cr-Cr coupling and more long-range Cr-Te-Cr play a crucial role in determining the magnetic exchange coupling and magnetic stability. Cr3+ has the d3 high-spin configuration occupying t2g orbital, and the super-exchange coupling through the Cr-Te-Cr bond angle is ferromagnetic, while the direct t2g-t2g coupling has antiferromagnetic characteristics, as shown in Figure B. In both the original and charge doped Cr2Ge2Te6 systems, the Cr-Te-Cr angles are greater than 90° (as shown in Table 1 of the revised manuscript), and the sign of all magnetic exchange constants is positive. Therefore, the magnetic stability of these systems is determined by super-exchange coupling. For the Cr-Te-Cr super-exchange coupling, although the antiferromagnetic coupling between Te and Cr atom is weak due to the tiny negative magnetic moment on Te atoms, this coupling is still essential. Therefore, it can be concluded that the weak antiferromagnetic coupling play a key role in magnetic stability to against the thermal fluctuation.

Figure B. Diagrams of Cr-Cr direct exchange coupling (a) and Cr-Te-Cr super-exchange coupling (b).

In the revised manuscript, we have added Figure B as Figure 7b-7c and added following sentences to address this point.

Page 12. “In Figure 2e&3c, it is evident that the spin orientation on Cr atom is antiparallel to that on Te atoms, disclosing a weak antiferromagnetic interaction between them. Although this antiferromagnetic interaction is very weak, it plays a fundamental role in stabilizing the ferromagnetic order in 2D Cr2Ge2Te6 and resisting the thermal fluctuation, as per the Goodenough-Kanamori rule79,80. In the Goodenough-Kanamori rule, the magnetic exchange constant is result of the competition between super-exchange and direct exchange coupling. The sign of the magnetic exchange constant depends heavily on the metal-ligand-metal angle. If the angle is greater than 90º, the magnetic exchange constant is dominated by the super-exchange coupling and its sigh is positive. If not, direct exchange coupling dominates the magnetic exchange coupling leading to a negative magnetic exchange constant. This rule is still applicable in newly discovered two-dimensional magnets, such as Cr2Ge2Te668, CrSTe81, CrI382, and MXenes83.”

Page 13. “In Cr2Ge2Te6, Cr3+ has the d3 high-spin configuration occupying t2g orbital, as shown in Figure 7b-7c. In DEI, direct t2g orbital overlapping between the intralayer adjacent Cr3+ ions accounts for the negative exchange constant, which is dominated primarily by the distance between nearest-neighbor Cr3+ ions. The Cr-Te-Cr SEC is caused by the overlap of t2g orbit on Cr3+ ion and p orbit on Te atom, depending on the magnetic moment on Te atom, Cr-Te bond length, and Cr-Te-Cr bond angle. Hence, the magnetic moment on Te atom, the Cr-Cr distance, the Cr-Te bond length and the Cr-Te-Cr bond angle in pristine and doped Cr2Ge2Te6 monolayer are exhibited in Table 1 to explore the origin of the enhancement on Tc. In both the pristine and charge doped Cr2Ge2Te6 systems, the Cr-Te-Cr bond angles are greater than 90° (as shown in Table 1) and the sign of all magnetic exchange constants is positive, while the Cr-Cr distance is almost unchanged with doping, indicating the insensitivity of antiferromagnetic DEI to the charge doping. Therefore, the magnetic stability of these systems is determined by SEC. For the Cr-Te-Cr SEC, although the antiferromagnetic coupling between Te and Cr atom is weak due to the tiny negative magnetic moment on Te atoms, this coupling is still essential. Therefore, it can be concluded that the weak antiferromagnetic coupling play a key role in magnetic stability to against the thermal fluctuation.”

[68] He, J., Ding, G., Zhong, C., Li, S., Li, D. & Zhang, G. J. Mater. Chem. C 7, 5084-5093, (2019).

[79] Kanamori, J. J. Phys. Chem. Solids 10, 87-98, (1959).

[80] Goodenough, J. B. Phys. Rev. 100, 564, (1955).

[81] Cui, Q., Liang, J., Shao, Z., Cui, P. & Yang, H. Phys. Rev. B 102, 094425, (2020).

[82] Pizzochero, M. & Yazyev, O. V. J. Phys. Chem. C 124, 7585-7590, (2020).

[83] Frey, N. C., Bandyopadhyay, A., Kumar, H., Anasori, B., Gogotsi, Y. & Shenoy, V. B. ACS Nano 13, 2831-2839, (2019).

Comment 4. Some details of the Monte Carlo simulation are missing: How large was the simulation box, were periodic boundary conditions assumed, and what was the strength of the applied external magnetic field (if any)? How many cycles were necessary to achieve equilibrium at each examined temperature? How large were temperature intervals?

Author Response: We thank the referee for this important comment. A 32×32×1 supercell was used in our Monte Carlo simulations, and 107 loops were implemented with Metroplis algorithm to achieve equilibrium at each examined temperature. During all simulations, the temperature increases from 2 to 300 K with step of ~5 K. It is noted that there is no external magnetic field applied in our Monte Carlo simulations.

In the revised manuscript, we have added following sentences to address this point.

Page 4. “In all Monte Carlo simulations, 32×32×1 supercell was used, and the temperature increases from 2 to 300 K with step of ~5 K. In every simulation, 107 loops were implemented with Metroplis algorithm to achieve equilibrium at each examined temperature. During all simulations, when there is no external magnetic field applied.”

Comment 5. It would be beneficial for the readers that the authors note that higher-level functionals that account for the electronic correlations are essential not only for obtaining accurate band structures and magnetic moments but also critical for predicting correct geometries, especially in doped systems: Pham et al. Critical role of Fock exchange in characterizing dopant geometry and magnetic interaction in magnetic semiconductors, Phys. Rev. B 89, 155110.

Author Response: We thank the referee for this important comment. According to your comment, we have downloaded this reference (PHYSICAL REVIEW B 89, 155110 (2014)) and read it carefully. In this reference, Pham et al. demonstrated the significance of the Fock exchange in accurately describing dopant geometry and the Co-O-Co super-exchange interaction in hydrogen-doped ZnO:Co system. In their study, the PBE method predicted the antibonding configuration of hydrogen (Co-H-Co) is more stable than the bond-centered configuration by 0.497 eV, while the HSE functional predicts the bond-centered hydrogen to be more stable than the Co-H-Co geometry of hydrogen by 0.126 eV. These opposing results between the two functionals can be owed to the nature of hydrogen bonding in the PBE and hybrid functionals. Meantime, the magnetic Co-O-Co super-exchange interaction is induced due to the crystal distortion caused by the doping of H and Co atoms in ZnO, so the Co-O-Co super-exchange interaction is also related strongly to the nature of hydrogen bonding in the PBE and hybrid functionals. In our work, there is no dopant atoms and new bonding in our charge doped Cr2Ge2Te6 systems, and the magnetic exchange coupling mainly relies on the distance between magnetic ions and the ligand-metal-ligand bond angle. Because the nature of bonding in the PBE and hybrid functionals is different, so the results obtained by the hybrid functionals, including the geometrical and magnetic exchange coupling, always is closer to the experimental results, compared with PBE+U method (J. Alloys Compd. 857, 157592, (2021)ï¼›Phys. Rev. B 101, 054429, (2020)). Furthermore, Joshi et al. (Polyhedron 176, 114194, (2020)) have proved that the difference between the magnetic exchange couplings obtained by the PBE+U and hybrid functionals is about ~2%. However, the consumption of hybrid functionals in memory and computational time is much larger than the PBE method of general gradient approximation (GGA), especially for magnetic materials. To balance the accuracy of calculation results and the resources required, we chose the PBE+U of GGA with spin polarization as exchange-correlation functional in all first-principles calculations for 2D magnets, as adopted in previous literatures (Nature Materials, 20, 818-825, 2021; PHYSICAL REVIEW B 98, 155148 (2018); PHYSICAL REVIEW B 104, 064443 (2021); Chinese Phys. Lett. 2020,  37 107506).

In the revised manuscript, we have added following sentences and references to address this point.

Page 3. “Early in 2014, Pham et al.42 demonstrated the significance of the Fock exchange in accurately describing dopant geometry and the super-exchange interaction in doped system. Generally, the results obtained by the hybrid functionals, including the geometrical and magnetic exchange coupling, is closer to the experimental results43,44, compared with the Perdew-Burke-Ernzerhof (PBE) method of general gradient approximation, because of the different nature of bonding in the PBE and hybrid functionals. In 2020, Joshi et al.45 proved that the difference between the magnetic exchange couplings obtained by the PBE and hybrid functionals is only about ~2%. However, the consumption of hybrid functionals in memory and computational time is much larger than the PBE method, especially for magnetic materials. To balance the accuracy and required resources of first-principles calculations, we chose the PBE of general gradient approximation with spin polarization as exchange-correlation functional46,47, as adopted in previous literatures48-51.”

[42] Pham, A., Assadi, M., Yu, A. & Li, S. Phys. Rev. B 89, 155110, (2014).

[43] Aras, M., Kılıç, Ç. & Ciraci, S. Phys. Rev. B 101, 054429, (2020).

[44] Gao, H., Li, M., Yang, Y. & Zhang, P. J. Alloys Compd. 857, 157592, (2021).

[45] Joshi, R. P., Phillips, J. J., Mitchell, K. J., Christou, G., Jackson, K. A. & Peralta, J. E. Polyhedron 176, 114194, (2020).

[46] Perdew, J. P., Burke, K. & Ernzerhof, M. Phys. Rev. Lett. 77, 3865, (1996).

[47] Kresse, G. & Hafner, J. Phys. Rev. B 55, 7539, (1997).

[48] Li, B., Wan, Z., Wang, C., Chen, P., Huang, B., Cheng, X., Qian, Q., Li, J., Zhang, Z., Sun, G. et al. Nat. Mater. 20, 818-825, (2021).

[49] Guo, Y., Liu, N., Zhao, Y., Jiang, X., Zhou, S. & Zhao, J. Chinese Phys. Lett. 37, 107506, (2020).

[50] Baidya, S., Yu, J. & Kim, C. H. Phys. Rev. B 98, 155148, (2018).

[51] Cheng, H.-X., Zhou, J., Wang, C., Ji, W. & Zhang, Y.-N. Phys. Rev. B 104, 064443, (2021).

Comment 6. The VASP LDA+U methods, specified by the LDAUTYPE key, are based on one of these two references. None were cited. If the authors used customized code, a note should be included in the paper:

  1. I. Liechtenstein, V. I. Anisimov, and J. Zaanen, Phys. Rev. B 52, R5467 (1995).
  2. L. Dudarev, G. A. Botton, S. Y. Savrasov, C. J. Humphreys, and A. P. Sutton, Phys. Rev. B 57, 1505 (1998).

Author Response: We thank the referee for this important comment. We have added “S. L. Dudarev, G. A. Botton, S. Y. Savrasov, C. J. Humphreys, and A. P. Sutton, Phys. Rev. B 57, 1505 (1998)” in our revised manuscript as Ref.[53] for LDAUTYPE key in the PBE+U method.

Reviewer 2 Report

Attached

Author Response

Response to Referee 2

We thank the referee for her/his insightful suggestions, which have been very helpful for us to improve the quality of our manuscript. We have revised the manuscript according to the referee’s suggestions. The changes in manuscript have been highlighted in the revised version. Below is our response to the comments.

Comment 1. Thermodynamics stability of the systems can be analysis by making the phase diagrams of the systems as discussed in this Ref. [Journal of Physical Chemistry C 118, 19625 (2014)], by computing the formation enthalpy/energy with respect to the convex hull rather than the elemental states. The authors can look up works by W. Sun, et al., Sci. Adv. 2016, 2 (11), S. Rubab, et al., Phys. Chem. Chem. Phys., 2021, 23, 19472 and Kirklin, S. et al., npj Computational Materials 2015, 1, 15010.

Author Response: We thank the referee for this important comment. We have downloaded and carefully read these four papers recommended by the reviewer. In Ref.[Journal of Physical Chemistry C 118, 19625 (2014)], formation energy was used to study the effect of hole doping caused by point defects on the thermodynamic stability of SrHfO3, while Ref.[Phys. Chem. Chem. Phys., 2021, 23, 19472] employed formation energy and investigated the structural stability of Nb and Tc doped Ba2CaMoO6 double perovskite oxide. The definition of formation energy Ef is as follows:

,

where Etot is the total energy of Cr2Ge2Te6 system at ferromagnetic ground state, while mCr, mGe and mTe are the chemical potential for Cr, Ge, and Te atoms, respectively. The calculated formation energies for Cr2Ge2Te6 monolayer with and without charge doping are presented in Table A. In Table A, we can find all of formation energies are negative, suggesting these systems in our manuscript are thermodynamically stable. Furthermore, doping with 0.3 electron per unit cell owns the lowest formation energy and the strongest thermodynamic stability.

Table A. Formation energy of Cr2Ge2Te6 monolayer with and without charge doping.

System

0.3 e

0.2 e

0.1 e

pristine

0.1 h

0.2 h

0.3 h

Formation energy (eV)

-14.62

-14.55

-14.43

-14.28

-14.05

-13.78

-13.46

In revised manuscript, we have added new column in Table 1 and following sentences to address this point.

Page 10-11. “Meantime, we also calculated the formation energies (Ef) by , as shown in Table 1. Etot is the total energy of Cr2Ge2Te6 system at ferromagnetic ground state, while mCr, mGe and mTe are the chemical potential for Cr, Ge, and Te atoms, respectively. It can be discovered in Table 1 that all of formation energies are negative, suggesting these systems in our manuscript are thermodynamically stable. In Table 1, 0.3e system owns the lowest formation energy, so its ferromagnetic order possesses the strongest thermodynamic stability, consistent with the largest Curie temperature in 0.3e system.”

Comment 2. Magnetic anisotropy energy (MAE) should be calculated. Because it is well established that for magnetic memory devices its value should be high for thermal stability of the devices while reducing their sizes as discussed in these references IEEE Magn. Lett. 2017, 8, 1-5. Moreover, effective anisotropy should be calculated, which is basically necessary parameter to check the device performance and should be explain in term of the second perturbation theory, see this Ref. [Phys. Rev. B 47, 14932 (1993).].  

Author Response: We thank the referee for this important comment. Magnetic anisotropic energy (MAE) is indeed one of the most important properties of magnetic materials (Phys. Rev. B 47, 14932 (1993); Journal of Magnetism and Magnetic Materials, 159, (1996), 337-341), especially when magnets are applied in nano magnetic memory devices (IEEE Magn. Lett. 2017, 8, 1-5). For two-dimensional magnetic materials, it is widely accepted that MAE arising from spin-orbit coupling (SOC) is one possibility to suppress thermal fluctuation. The existence of magnetic anisotropy can reduce the total energy of the system and result in the presence of a spin-wave excitation gap, thus suppressing the effect of thermal fluctuations (Nature 546, 265-269, (2017); Nat. Nanotechnol. 14, 408-419, (2019)). Overall, the existence of magnetic anisotropy can make the spin orientation more stable, which improves the stability of magnetic ordering to resist thermal fluctuations. According to this comment from referee, we have calculated the magnetic anisotropic energies in Cr2Ge2Te6 monolayers with and without charge doping. The out-of-plane MAE is only 0.23 meV in pristine Cr2Ge2Te6 monolayer, which is ~3.3% of the magnetic exchange constant (6.874 meV) and consistent with the results of Xu et al (npj Computational Materials (2018) 4:57). and Zhang et al (Phys. Rev. B 100, 224427 (2019)). After 0.3 electron doping, MAE transfers from the out-of-plane to the in-plane direction and increases to 0.59 meV, keeping only ~5.7% of magnetic exchange constant (10.202 meV). Meantime, 0.3 hole doping results in an out-of-plane MAE of 0.42 meV. It can be found that MAE in Cr2Ge2Te6 monolayer is one order of magnitude smaller than the magnetic exchange constant, regardless of whether charge doping is applied or not.

In the revised manuscript, we have added following sentences and references to address this point.

Page 7-8. “It has been mentioned above that the out-of-plane magnetic anisotropic energy of Cr2Ge2Te6 monolayer is only about 3%~5% of the nearest-neighbouring exchange constant60,61, thus always is neglected in the estimation of Curie temperature. However, whether charge doping increases the magnetic anisotropy in Cr2Ge2Te6 monolayer or not? For magnetic materials, Magnetic anisotropic energy (MAE) is one of the most important properties of magnetic materials64,65, especially when magnets are applied in nano magnetic memory devices66. We have calculated the magnetic anisotropic energies in Cr2Ge2Te6 monolayers with and without charge doping. The out-of-plane MAE is only 0.23 meV in pristine Cr2Ge2Te6 monolayer, which is ~3.3% of the magnetic exchange constant (6.874 meV) and consistent with the results of Xu et al.61 and Zhang et al59. After 0.3 electron doping, MAE transfers from the out-of-plane to the in-plane direction and increases to 0.59 meV, keeping only ~5.7% of magnetic exchange constant (10.202 meV). Meantime, 0.3 hole doping results in an out-of-plane MAE of 0.42 meV. It can be found that MAE in Cr2Ge2Te6 monolayer is always one order of magnitude smaller than the nearest-neighbouring magnetic exchange constant, regardless of whether charge doping is applied or not. Therefore, we did not pay too much attention on it in this work.”

[59] Zhang, B. H., Hou, Y. S., Wang, Z. & Wu, R. Q. Phys. Rev. B 100, 224427, (2019).

[60] Fang, Y., Wu, S., Zhu, Z.-Z. & Guo, G.-Y. Phys. Rev. B 98, 125416, (2018).

[61] Xu, C., Feng, J., Xiang, H. & Bellaiche, L. NPJ Comput. Mater. 4, 57, (2018).

[64] Wang, D.-s., Wu, R. & Freeman, A. Phys. Rev. B 47, 14932, (1993).

[65] Wang, X., Wang, D.-s., Wu, R. & Freeman, A. J. Magn. Magn. Mater. 159, 337-341, (1996).

[66] Peng, S., Kang, W., Wang, M., Cao, K., Zhao, X., Wang, L., Zhang, Y., Zhang, Y., Zhou, Y., Wang, K. L. et al. IEEE Magnetics Letters 8, 1-5, (2017).

Comment 3. For practical implementation, it is necessary to check the robustness of half-metallic state of the doped systems via biaxial strain, specifically 0.3 electrons per unit cell doped one.   

Author Response: We thank the referee for this important comment. We have calculated the band structures of doped Cr2Ge2Te6 monolayer with 0.3 electrons, as shown in Figure C(a) where the spin-up and spin-down channels are represented by the red and blue lines, respectively. Obviously, the 0.3 electron doped Cr2Ge2Te6 monolayer (0.3e system) shows a half-metallic behaviours. To test the robustness of this half-metallic state in doped system, we also calculated the band structures of 0.3e system under 2% biaxial lattice expansion and compression in Figure C(b-c). When biaxial strain is applied, it can be found that the conduction band minimum (CBM) remains almost unchanged, while the valence band maximum (VBM) reduces with expansion and elevates with compression, but the half-metallic state in doped system is not affected. This phenomenon indicates the half-metallic state in doped system is robust.

Figure C. (a) Band structure of 0.3 electron doped Cr2Ge2Te6 monolayer (0.3e system), and band structures of 0.3e system under 2% biaxial expansion (b) and compression (c). The spin-up and spin-down channels are represented by the red and blue lines, respectively.

In the revised manuscript, we have added Figure C as Figure 6 and added following sentences to address this point.

Page 11. “Besides, we have calculated the band structures of doped Cr2Ge2Te6 monolayer with 0.3 electrons, as shown in Figure 6a where the spin-up and spin-down channels are represented by the red and blue lines, respectively. Obviously, the 0.3 electron doped Cr2Ge2Te6 monolayer (0.3e system) shows a half-metallic behaviours. To test the robustness of this half-metallic state in doped system, we also calculated the band structures of 0.3e system under 2% biaxial lattice expansion and compression in Figure 6b-6c. When biaxial strain is applied, it can be found that the conduction band minimum (CBM) remains almost unchanged, while the valence band maximum (VBM) reduces with expansion and elevates with compression, but the half-metallic state in doped system is not affected. This phenomenon indicates the half-metallic state in doped system is robust.”

Comment 4. Authors claim that Tc improved up to 44.96% for 0.3 electrons per unit cell doping, so what is the physical phenomena behind this?

Author Response: We thank the referee for this important comment. We cannot understand the meaning of this expression ‘the physical phenomenon behind the increase in Tc. The physical meaning of magnetic phase transition temperature is the critical temperature at which magnetic materials transition from a magnetic ordered state (ferromagnetic or antiferromagnetic state) to a magnetic disordered state (paramagnetic state). In recent years, 2D magnetic materials have received much attention due to their fascinating properties, but the magnetic phase transition temperature of most 2D magnetic materials is far below 100K, such as ~21 K for seven-layer MnBi2Te4 (National Science Review 7 (8), 1280-1287 (2020)), ~45 K for CrI3 (Nature 546, 270-273, (2017)) and ~89 K for FePS3 (J. Phys. D: Appl. Phys. 54, 314001, (2021)). We find that doping with 0.3 electron can increase the magnetic phase transition temperature of Cr2Ge2Te6 from 85 K to 123 K. In experiments, charge doping can be realized by applying a gate voltage, based on an electric transistor device. Therefore, the corresponding physical phenomena is that the Curie temperature of Cr2Ge2Te6 monolayer increases with positive gate voltage. Our results suggest the Curie temperature of 2D magnetic material Cr2Ge2Te6 can be effectively regulate by applying gate voltage in practical applications, which is of significance for the development of 2D magnetic materials and their applications in future spin electronic devices.

Comment 5. GGA+U method is used as an exchange correlation functional. However, I would like to suggest that the HSE06 functional provides much better results and understanding for such kind of complex systems.

Author Response: We thank the referee for this important comment. Because the nature of bonding in the PBE and hybrid functionals is different, so the results obtained by the hybrid functionals, including the geometrical and magnetic exchange coupling, always is closer to the experimental results, compared with PBE+U method (Phys. Rev. B 101, 054429, (2020); J. Alloys Compd. 857, 157592, (2021)). Furthermore, Joshi et al. (Polyhedron 176, 114194, (2020)) have proved that the difference between the magnetic exchange couplings obtained by the PBE+U and hybrid functionals is about ~2%. However, the consumption of hybrid functionals in memory and computational time is much larger than the PBE method of general gradient approximation (GGA), especially for magnetic materials. To balance the accuracy of calculation results and the resources required, we chose the PBE+U of GGA with spin polarization as exchange-correlation functional in all first-principles calculations for 2D magnets, as adopted in previous literatures (Nature Materials, 20, 818-825, 2021; PHYSICAL REVIEW B 98, 155148 (2018); PHYSICAL REVIEW B 104, 064443 (2021); 2020 Chinese Phys. Lett. 37 107506).

In the revised manuscript, we have added following sentences and references to address this point.

Page 3. “Early in 2014, Pham et al.42 demonstrated the significance of the Fock exchange in accurately describing dopant geometry and the super-exchange interaction in doped system. Generally, the results obtained by the hybrid functionals, including the geometrical and magnetic exchange coupling, is closer to the experimental results43,44, compared with the Perdew-Burke-Ernzerhof (PBE) method of general gradient approximation, because of the different nature of bonding in the PBE and hybrid functionals. In 2020, Joshi et al.45 proved that the difference between the magnetic exchange couplings obtained by the PBE and hybrid functionals is only about ~2%. However, the consumption of hybrid functionals in memory and computational time is much larger than the PBE method, especially for magnetic materials. To balance the accuracy and required resources of first-principles calculations, we chose the PBE of general gradient approximation with spin polarization as exchange-correlation functional46,47, as adopted in previous literatures48-51.”

[42] Pham, A., Assadi, M., Yu, A. & Li, S. Phys. Rev. B 89, 155110, (2014).

[43] Aras, M., Kılıç, Ç. & Ciraci, S. Phys. Rev. B 101, 054429, (2020).

[44] Gao, H., Li, M., Yang, Y. & Zhang, P. J. Alloys Compd. 857, 157592, (2021).

[45] Joshi, R. P., Phillips, J. J., Mitchell, K. J., Christou, G., Jackson, K. A. & Peralta, J. E. Polyhedron 176, 114194, (2020).

[46] Perdew, J. P., Burke, K. & Ernzerhof, M. Phys. Rev. Lett. 77, 3865, (1996).

[47] Kresse, G. & Hafner, J. Phys. Rev. B 55, 7539, (1997).

[48] Li, B., Wan, Z., Wang, C., Chen, P., Huang, B., Cheng, X., Qian, Q., Li, J., Zhang, Z., Sun, G. et al. Nat. Mater. 20, 818-825, (2021).

[49] Guo, Y., Liu, N., Zhao, Y., Jiang, X., Zhou, S. & Zhao, J. Chinese Phys. Lett. 37, 107506, (2020).

[50] Baidya, S., Yu, J. & Kim, C. H. Phys. Rev. B 98, 155148, (2018).

[51] Cheng, H.-X., Zhou, J., Wang, C., Ji, W. & Zhang, Y.-N. Phys. Rev. B 104, 064443, (2021).

Round 2

Reviewer 1 Report

The extent of the revisions are impressive, and the paper can be accepted.

Reviewer 2 Report

All the quires has been explained by the authors, therefore, manuscript is ready for publication.